Dynamic analysis and control of a rice-pest system under transcritical bifurcations

Mandal Sajib Sajib.Mandal@student.uts.edu.au 1
Oberst Sebastian 1
Biswas Md. Haider Ali 2
Islam Md. Sirajul 3
1 Centre for Audio, Acoustics and Vibration, University of Technology Sydney , Sydney , Australia
2 Mathematics Discipline, Khulna University , Khulna , Bangladesh
3 Department of Mathematics, Bangabandhu Sheikh Mujibur Rahman Science and Technology University , Gopalganj , Bangladesh
Shang Yilun
Electronic publication date: 2023 Oct 10
Publication date: 2023
Volume: 11
Electronic Location ID: e16083
Received 2022 Sep 26; Accepted 2023 Aug 21
Copyright: ©2023 Mandal et al.
Copyright year: 2023
Copyright holder: Mandal et al.
License: This is an open access article distributed under the terms of the Creative Commons Attribution License, which permits unrestricted use, distribution, reproduction and adaptation in any medium and for any purpose provided that it is properly attributed. For attribution, the original author(s), title, publication source (PeerJ) and either DOI or URL of the article must be cited.
License URL: https://creativecommons.org/licenses/by/4.0/

Keywords: Optimal control, Decision model, Integrated pest management, Nonlinear dynamics, Lotka-Volterra model, Predator-prey

Funding: The authors received no funding for this work.

==============================
A decision model is developed by adopting two control techniques, combining cultural methods and pesticides in a hybrid approach. To control the adverse effects in the long term and to be able to evaluate the extensive use of pesticides on the environment and nearby ecosystems, the novel decision model assumes the use of pesticides only in an emergency situation. We, therefore, formulate a rice-pest-control model by rigorously modelling a rice-pest system and including the decision model and control techniques. The model is then extended to become an optimal control system with an objective function that minimizes the annual losses of rice by controlling insect pest infestations and simultaneously reduce the adverse impacts of pesticides on the environment and nearby ecosystems. This rice-pest-control model is verified by analysis, obtains the necessary conditions for optimality, and confirms our main results numerically. The rice-pest system is verified by stability analysis at equilibrium points and shows transcritical bifurcations indicative of acceptable thresholds for insect pests to demonstrate the pest control strategy.

Introduction

A variety of types of rice are cultivated in the world depending on environmental factors and water availability. Oryza sativa is globally cultivated, especially in Asia (Vaughan, Lu & Tomooka, 2008), Oryza glaberrima in African rice (Judith, 2004), but several species of the genera Zizania and Porteresia known as wild rice such as Fritillaria camschatcensis are mainly cultivated in South Asia, North America, and China (IRRI, 2014). Rice has a three-phase crop cycle: vegetative, reproductive and ripening, and takes about 100 to 210 days to yield crop (Zadoks, Chang & Konzak, 1974). The growth of paddy plants varies with environmental factors such as temperature, air humidity and water level, rice varieties, pest infestation, and amount of pesticides used (IRRI, 2020; Zadoks, Chang & Konzak, 1974). The average harvesting period in temperate countries ranges from 130 to 150 days (Zadoks, Chang & Konzak, 1974). Some rice varieties which are photoperiod sensitive and planted non-shallow water take 150 to 210 days to grow crop (Zadoks, Chang & Konzak, 1974). Due to pest infestations, overuse of pesticides, and also global effects such as climate change, the growth of rice can be impeded (IRRI, 2020). Yet, most of the rice species are cultivated twice a year, and farmers need time to prepare the cultivation lands and seeds for the next season’s rice yield. The time interval takes generally 10 to 30 days depending on farming equipment and cultivation areas (IRRI, 2020). Rice species which require about 7 months to provide yield are only cultivated once a year (IRRI, 2020).

Invertebrate species can be found in the paddy field varying with location, environmental factors, density and development stages of the rice plant. Among the hundreds of species, several identified species have classified as pests to the rice crop, and which significantly damage paddy plants or reduce yield and are able to directly or indirectly transmit diseases. One of these critical species is the brown planthopper (Khush, 1999), another one is the rice gall midge (Benett et al., 2004), or also several species of stemborers (IRRI, 2014), and rice bugs (Jahn et al., 2004), notably found within the genus Leptocorisa (Jahn et al., 2004), as well as defoliators such as leafrollers (rice hispas), cf (Gurr et al., 2012). Rice leafrollers (Cnaphalocrocis medinalis) are harmful at the larval stage with a single larva able to consume up to 40% of a rice leaf and making its tip spiral into an insect bud (Gurr et al., 2012). The brown planthopper (Nilaparvata lugens) damages rice directly through feeding and also by transmitting viruses (Khush, 1999). The larvae of stemborers (Scirpophaga incertulas and Scirpophaga innotata), damage paddy plants, especially their leaves (IRRI, 2014). Chilo suppressalis attacks almost all plant parts ranging from the leaf to the root (IRRI, 2014). The rice bug Oebalus spp. attacks particularly the rice plant’s more developed panicle stages (Jahn et al., 2004).

With the increase in the annual rice yield, the number of annual production and losses gradually increases as shown in Fig. 1, which depicts the annual losses from 1960 (∼10.6 millions of metric tons (MMT)) to 2013 (∼41.05 MMT). On the other hand, MMT of rice are lost annually only due to pest infestations amounting to about 37% of the annual production in the world (FAO, 2003; IRRI, 2020). Several environmental factors and human-made issues are responsible for continuing annual losses of rice, such as climate change, viral and bacterial diseases, fungal infections, deforestation, environmental pollution, excessive use of pesticides, and farmers’ less attention to rice crops. However, the increase of pest density in cultural areas due to deforestation, urbanisation and industrialisation is considered as the greatest problem (Zhao et al., 2021). The common major factors responsible for annual global losses of rice are described in Fig. 1. Due to global warming, the temperature-sensitive rice species slow down productive capacity and increase the adaptation and reproductive capacity across a range of pest species by causing distributional shifts that also reduce rice production (Parmesan, 2006; Sánchez-Guillén et al., 2016; McCulloch & Waters, 2023). In addition, farmers may be responsible for the low rice production and rice losses due to concentrating on other profitable crops, paying less attention to surrounding factors that may adversely affect rice, and cultivating other crops after the rice season (Nature & Farming, 2015).

Figure 1 Annual production and losses of rice from 1960–2013 and the most common factors responsible for the rapid increase in annual global losses of rice over the years.

(Left) Annual production and losses of rice from 1960–2013 (FAO, 2021) represented under a logarithmic scale with MMT measurement unit. In comparison to production, annual losses increased over the years and their absolute number is very high, and (right) the most common factors responsible for the rapid increase in annual global losses of rice over the years. The increasing density of pests and excessive use of pesticides in paddy fields are the major causes of the losses of rice. Only the factors identified in red frames are considered in this study.

Chemical controls (pesticides) are conventionally used in modern agriculture for instant pests control. Neonicotinoids or glyphosate, DDT, BHC, and 2,4-D are commonly used (Inao et al., 2008) yet their applications are increasingly scrutinised due to potential long-lasting negative effects on the environment, especially insect populations—essential for crop pollination (Marinelli, 2005). Besides, the overuse of pesticides and insecticides seriously hampers food quality and adversely impacts aquatic ecosystems (Damalas & Eleftherohorinos, 2011). To avoid problems with chemicals, alternative controls have been sought including cultural methods, biological controls, or combinations thereof (Sun, Zhang & Tian, 2017; Peshin & Dhawan, 2009). Cultural methods consist of crop rotation, soil enhancement, healthy crop maintenance, and field sanitation. This means in practice the removal of diseased plants, proper irrigation, and the active encouragement of certain predatory species (biological controls), which does not require additional artificial chemical substances. Cultural methods are hence less invasive and contribute naturally to increased productivity (Peshin & Dhawan, 2009; Nature & Farming, 2015). Biological controls constitute effective methods including natural enemies of insect pests as control agents such as parasitoids (e.g., Diapriidae for flies) (Garay et al., 2015), predators (e.g., Ophionea nigrofasciata) (Ooi & Shepard, 1994), genetic sterilisation (Alphey & Bonsall, 2018), pathogens (viral infections) (Bhattacharyya & Bhattacharya, 2006), or the exploitation of competitor relationships (e.g., competition for preying between the protist Didinium and Paramecium) (Flint & Dreistadt, 1998). Yet, biological controls are expensive to impose, and require a long-term process and a pest density at manageable thresholds (Tang & Cheke, 2008).

Optimised integrated pest management (IPM) strategies (including cultural methods, biological methods, and pesticides) aim to increase crop yield by mitigating pest infestation (Nieto & O’Regan, 2009). Biological food web models have been developed considering either biological controls (impulsive release of natural enemies) or pesticides (Changguo, Yongzhen & Xuehui 2009; Liu, Xu & Kang, 2013; Jatav & Dhar, 2014; Song, Wang & Jiang, 2014). Sun, Zhang & Tian (2017) formulated an optimal control strategy approaching a mathematical predator–prey model considering biological and chemical controls, with the latter acting faster. Huang et al. (2019) analyzed a predator–prey model to describe how natural enemies reduce agricultural pests and to investigate the impact of pesticides on pests management. Tang, Tang & Cheke (2010) showed by incorporating different regulatory methods that factors such as residual effects and spray times of pesticides, or release time of natural enemies have a significant impact on pests population. Besides, biological controls imply using predator–prey models with a periodic release of predators (Mailleret & Grognard, 2009), periodic application of pesticides to the pest species (Nundloll, Mailleret & Grognard, 2010), or pests becoming infected through bacterial strains or viruses (Chunyan & Daqing, 2013). However, IPM strategies focusing only on biological controls are limited and usually require knowledge of several input factors such as host-parasite ratios, start and end points of population densities, parasite population or insecticide dosage as well as release duration, and levels of parasitism or host-feeding (Tang & Cheke, 2008). Whilst previous works mainly focused on predator–prey relationships and pest control using biological controls and pesticides, none developed an optimal control model that can increase rice yield by controlling rice pests and minimize environmental pollution as well. In particular, how and when pesticides should be applied so that adverse effects of pesticides on the environment can be avoided have not been addressed in more detail.

Thus, to contribute to closing this research gap, a rice-pest-control model describing effective control techniques and a decision model of proper application strategies with minimum adversity of pesticides on the environment are investigated for the first time. The decision model adopts two controls, using cultural methods and pesticides, to determine the start and stop instances of each technique. We then formulate a rice-pest-control model by introducing the decision model and its control techniques, which are then transformed into an optimal control problem (OCP) and obtained the necessary conditions for optimality of the OCP using Pontryagin’s maximum principle in terms of the Hamiltonian. The rice-pest-control model is developed on a simple predator–prey model of a rice-pest system and partially describes the agricultural control system due to avoiding some factors that have impacts on the cultivation and production of rice, the limitations and future works of this study are finally discussed.

Methods

Study area

This study has been conducted globally without being confined to a specific region because food security and pest management are global issues. It is not possible to reduce the annual loss of rice and control pest infestation by adopting certain control strategies in certain areas of a country. Here, we consider two control strategies, cultural methods and chemical control, to increase the annual production of rice by controlling the pests in the paddy field. Cultural methods which consist of natural controls such as soil rotation, crop variation, and natural enemies are used as the first and foremost control strategy because of having natural capacity of increasing the production of rice and controlling rice pests. On the other hand, chemical controls consisting of various pesticides such as neonicotinoids, glyphosate, DDT, BHC, and 2,4-D are used only for emergency situations because of having adverse effects on the environment and crop quality. Here, we do not define any specific pesticide because pesticides can vary from pest to pest. Besides, the cultivation areas/lands are decreasing due to deforestation and urbanization, so the total cultivation areas are not constant and change every year. To avoid the effect of this contradiction in this study, the annual production of rice is considered and calculated in metric tons per hectare area (Mt/h).

Lotka–Volterra model in an optimal control problem setting

The general form of the Lotka–Volterra model can be written in terms of a pair of autonomous differential equations (1) x ˙=xfx,yy ˙=ygx,y

with x and y being the prey and predator populations and f(x, y) and g(x, y) being the species’ growth rates.

The optimal control problem (OCP) of the Lotka–Volterra model is to find control variable sets to optimize (minimize or maximize) a specific objective function (Lenhart & Workman, 2007), subject to constraints on the state variables. To define how an OCP is concerned with the state and the control variables, consider x(t) and u(t) are these two variables of an OCP respectively, x(t) satisfies (2) x ˙t=gt,xt,ut

with g being continuously differentiable. An OCP can be described in the following way. (3) Maximize or minimizeJx,u= ∫abLt,xt,utdtSubject tox ˙t=gt,xt,uta.e.t∈a,but∈U,∀t∈a,bxa∈x0andxb∈R+is free

with u(t) and x(t) both being piecewise C1 differentiable with t in [a, b] being the time interval where a, b ∈ ℝ+ and a < b. In the problem, u(t) belongs to a certain space U that may be a piecewise continuous function or a space of measurable function which satisfies all the constraints of the problem. Therefore, (x∗, u∗) constitutes the optimal solution of the OCP in case the costs can be minimized overall admissible processes (Lenhart & Workman, 2007).

Dynamical modelling in form of a hybrid natural/chemical IPM strategy

Portions of this text were previously published as part of a preprint (Mandal et al., 2022).

Assumption and formulation of the optimal control problem

Wiping out the entire pest population from the agricultural field is impossible, and an attempt can be unsafe, expensive, and may lead to a rebound in pest numbers (IPM, 2021). To control the pest population, we first apply cultural methods that are safe, cheap, and easily applicable. We then model the application of chemical controls (pesticides) only when the unacceptable action thresholds are crossed (IPM, 2021). The application of pesticides depends on the species of pest and their density.

An important question is the determination of the point in time when to apply pesticides and for how long it should continue. Let u1 denote cultural methods which are used incessantly throughout the entire cultivation period because they are natural and safe. Let u2 denote chemical controls (pesticides) which are used only when the pest level crosses the acceptable threshold. The controls u1 and u2 both take values between 0 and 1. Here, u1, u2 = 0 denote that no control is applied and u1, u2 = 1 indicate controls are applied with full effort. Let AT denote the unacceptable threshold (or action threshold) to determine the application of pesticides to control the pest population, and ACT represents the interval in which pesticides should be applied. The unacceptable threshold can be defined as: AT = μΔ, where µdenotes an unacceptable pest population per unit area and Δ represents the total area of a field. The threshold can depend on the pest variety, pest size, and site/region (IPM, 2021). For an increase in crop damaging pests, pesticides for their control are applied which may cause additional environmental impact through spreading via wind or rainwater into nearby ecosystems. Let D(x2, u2, E) denote the damage function of the nearby ecosystems which is potentially subject to excessive pesticide application, with x2 being the pest population and E representing the environmental pollution. E is defined in the interval [0, 1] with E = 1 denoting that the pollution reaches the maximally acceptable level (E = 0 denotes zero pollution). The damage function, D, is increased in the rise of E, but the application of u2 is decreased in the increase of D, where D ∈ [0, 1] (Lichtenberg & Zilberman, 1986; Waterfield & Zilberman, 2012). D = 1 represents the maximally acceptable damage threshold, meaning the application of pesticides should be stopped i.e., u2 = 0. Conversely, D = 0 denotes that there is no damage, meaning pesticides can be applied with full effort i.e., u2 = 1. Here E is proportionally increased with the application of u2. If the pest species population crosses the action threshold, pesticides are employed i.e., u2t∈0,1whenx2t>AT which increases the environmental pollution i.e., E ∈ (0, 1]. If E → 1 as u2 → 1 which leads D → 1 that means the use of pesticides should be stopped immediately i.e., u2 = 0. To reflect this, the following decision model has been developed: (4) FD,t=u1t∈0,1u2t=0,∀x2t≤ATu1t,u2t∈0,1u2t=1−Dx2t,u2t,Et,∀x2t>AT

where D is defined as, Dx2,u2,E=D=0whenu2,E=0,∀x2t≤AT0<D<1whenu2,E∈0,1D=1whenu2,E=1,∀x2t>AT.

A decision-making diagram of the model is expressed by Eq. (4) which describes the best time for the applications of pesticides is presented in Fig. 2. According to Fig. 2, when x2 crosses the AT line, u2 is applied. Due to the applications of u2, the growth rate of x2 gradually slows down. When u2 is applied at full scale (u 2 =1), the growth rate of x2 starts decreasing after reaching a stable situation. Because u 2 =1, D reaches the maximum level i.e., D =1; the use of pesticides is stopped (u 2 =0) to mitigate the adverse effect of pesticides on the environment and nearby ecosystems.

Figure 2 A decision-making diagram of the decision model Eq. (4) that describes the ideal timing for the application of pesticides.

Here, D = 1 represents the acceptable damage threshold reaching a maximum of still acceptable pollution. AT denotes the action threshold to determine the use of pesticides to control the pest population, and ACT represents the interval in which pesticides should be applied. When the density of the pest population crosses the AT line, pesticides are applied. Due to the use of pesticides, the growth rate of pests gradually slows down. When the pesticides are used at full scale (u2 =1), the growth rate of pests starts decreasing after reaching a stable situation. Because u2 = 1, the acceptable damage threshold reaches the maximum level i.e., D = 1; the use of pesticides is stopped (u2 = 0) to limit the environmental pollution (defined in the 4th row of Eq. (4)).

A control model is formulated on a developed rice-pest model, expressed by Eq. (S4) presented in Supplemental Information 1, where cultural methods and chemical controls are applied as two control variables aiming to minimize the density of rice pests. The dynamic relationships between the annual production of rice and the pest population under control are described below:

i. When cultural methods are applied during cultivation, the production rate of rice increases rapidly. Since cultural methods can directly increase the production of rice and the use of cultural methods does not depend on the density of rice pests, let u1x1 be the increment in the production of rice due to cultural methods. The first equation of the rice-pest system (S4) can be represented by including control as (5) ddtx1t=α1−β1x2tx1t−d1x12t+u1tx1t

here, x1(t) represents the annual production of rice per unit area (Mt/h), α1 shows the reproduction rate of rice, β1 represents the loss rate of x1(t) due to the consumption of pests, d1 presents the decrease rate due to intraspecific competition in species x1(t) due to natural causes that are not related to pests e.g., viral infections, droughts or floods (Bazykin, 1976; Liu & Jiang, 2021). Please note that pesticides do not directly increase the production of rice, but directly control the level of pest species causing an indirect increase of rice production. Since the term “ β1x2x1” in Eq. (5) represents the impact of pest density, no additional term is required to present the effect of pesticides on the rice production.

ii. When cultural methods are applied in rice cultivation, several rice pests die off because of soil rotation and the presence of predators. Therefore, the cultural methods decline the density of rice pests and let u1x1x2 be the declining number of pests due to the adoption of cultural methods. On the other hand, when emergency situations, pesticides are applied according to Eq. (4), which significantly reduces the pest population. Since the application of pesticides depends on the density of insect pests, let u2x1x2 be the decline in the density of pest population after the use of pesticides. Hence, the second equation of the rice-pest system (S4) can be represented considering the decision model Eq. (4) as in the following: (6) ddtx2t=β2x1t−α2x2t−d2x22t−FD,tx1tx2t

here, x2(t) represents the density of rice pests at time t, β2 shows the energy gain rate of pest population by consuming rice, α2 represents the decline rate of the pest’s population proportionally with the decline of rice production, d2 shows the decrease rate due to intraspecific competition between x2(t) due to natural causes that not related to x1(t), e.g., viral infection and heavy rains (Bazykin, 1976; Yang, 2020). Here, the term F(D, t)x1x2 corresponds to u1x1x2 and u2x1x2 since the decision model Eq. (4), defined by F(D,t), decides the application of the control variables u 1 and u 2.

The modified rice-pest system (S4) under controls and decision model Eq. (4) can be represented by arranging Eqs. (5) and (6) as in the following: (7) ddtx1t=α1−β1x2tx1t−d1x12t+u1tx1tddtx2t=β2x1t−α2x2t−d2x22t−FD,tx1tx2txt=x1t,x2t,∀txt=x10,x20,whent=0

Therefore, the system defined by Eq. (7) represents the rice-pest-control model, in Lotka–Volterra form complemented by a decision model Eq. (4). The decision model (Eq. (4)) controls the results of the rice-pest-control model Eq. (7) and mitigates the adversity of pesticides by controlling their use.

For more details, please look at Supplemental Information 1 for rice-pest system (S4) formulation and analysis, and Supplemental Information 2 for transcritical bifurcation analysis for the rice-pest system (S4).

The characteristics of the controls are represented in the following measurable control set. (8) U=u1t,u2t:0≤uit≤1,i=1,2att∈0,T

where T is a preselected period for applied controls. The objective function of the control model Eq. (7) becomes (9) MinimizeJx,u= ∫0Tx2t+A2u12+B2u22dt

The optimal control model which approximates model Eq. (7) can be represented (Lenhart & Workman, 2007) as: (10) MinimizeJx,u= ∫0Tx2t+A2u12+B2u22dtSubject tox′=gt,xt,utut∈U,∀t∈0,Tx0=x0

where xt=x1tx2t, gt,xt,ut=α1−β1x2tx1t−d1x12t+u1tx1tβ2x1t−α2x2t−d2x22t−FD,tx1tx2t and ut=u1tu2t; A and B are used for cost balancing weight parameters for the control variables u1 and u2, respectively; the function g is continuously differentiable; and functions u(t) and x(t) are piecewise continuous differentiable. In this problem, u(t) belongs to a certain space U that may be a piecewise continuous function or a space of measurable functions which satisfy all constraints of the problem. The main goal of the objective function is to increase the annual rice yield by minimizing the pest population by simultaneously considering the controls with the lowest costs. A schematic diagram of the rice-pest-control system Eq. (7) is shown in Fig. 3 to illustrate the dynamic behaviour of the species under control.

Figure 3 Schematic diagram of the rice-pest-control system Eq. (7) describes the rice-pest system (S4) under control.

The diagram also shows that the control strategies, cultural methods and chemical control (pesticides), increase rice production and reduce corresponding pest populations.

Characterization of the optimal control

To estimate the necessary conditions for the optimality of the optimal control problem Eq. (10), Pontryagin’s maximum principle has been imposed in terms of the Hamiltonian H(t) defined as Lenhart & Workman (2007) Ht,x,u,λ=Lt,x,u+∑i=12λigt,x,u=x2t+A2u12t+B2u22t+λ1α1−β1x2tx1t−d1x12t+u1tx1t+λ2β2x1t−α2x2t−d2x22t−FD,tx1tx2t

where λi, i = 1, 2 is the co-state variable which satisfies the following adjoint equations,

dλ1dt=−dHt,x,u,λdx1t =−λ1α1−β1x2t−2d1x1t+u1t−λ2β2x2t−FD,tx2t, and

dλ2dt=−dHt,x,u,λdx2t =−1+λ1β1x1t+λ2α2−β2x1t+2d2x2t+FD,tx1t

as well as the transversality conditions λ1(T) = 0 and λ2(T) = 0.

Now, to obtain the optimal solution of the controls, Theorem 3.1, and Theorem 3.2 must be proven as shown below by applying Pontryagin’s maximum principle.

Theorem 3.1. The control variables for the acceptable damage threshold attain the optimal solutions

u1∗,u2∗=max0,min1,λ2x1∗x2∗−λ1x1∗A,0 for which the objective function J over U is minimized.

Proof At the acceptable damage threshold, there is no use of pesticides i.e., u2 = 0. Therefore, let’s differentiate the Hamiltonian (H) with respect to the control variable u1 only, then it becomes as (11) dHdu1=Au1+λ1x1−λ2x1x2

By applying the conditions of optimality in Eq. (11), the characterization of the control variable u1

i. when dHdu1>0 then u1>λ2x1x2−λ1x1A but for the minimization problem u1 = 0

ii. when dHdu1=0 then u1=λ2x1x2−λ1x1A

iii. when dHdu1<0 then u1<λ2x1x2−λ1x1A but for the minimization problem u1 = 1

Therefore u1∗=1whendH/du1<0λ2x1x2−λ1x1AwhendH/du1=00whendH/du1>0, which can be written in the following compact form

u1∗= max0,min1,λ2x1∗x2∗−λ1x1∗A.

Since u2 = 0 at the acceptable threshold, the compact form of u2 is u2∗ = 0. Then, the optimal solutions of the control variables for the acceptable threshold are

u1∗,u2∗=max0,min1,λ2x1∗x2∗−λ1x1∗A,0,

hence, it completes the proof.

Theorem 3.2. The control variables for the action threshold attain the optimal solutions

u1∗,u2∗=max0,min1,λ2x1∗x2∗−λ1x1∗A,max0,min1,λ2x1∗x2∗B for which the objective function J over U is minimized.

Proof: For the action threshold, the decision model is fully active i.e., F(D, t) = u1(t) + u2(t). Therefore, let’s differentiate the Hamiltonian (H) with respect to the control variables u1 and u2, then it becomes as

(12) dHdu1=Au1+λ1x1−λ2x1x2

(13) dHdu2=Bu2−λ2x1x2

By applying the conditions of optimality in Eq. (12), the characterization of the control variable u1

i. when dHdu1>0 then u1>λ2x1x2−λ1x1A but for the minimization problem u1 = 0

ii. when dHdu1=0 then u1=λ2x1x2−λ1x1A

iii. when dHdu1<0 then u1<λ2x1x2−λ1x1A but for the minimization problem u1 = 1

Therefore u1∗=0whendH/du1>0λ2x1x2−λ1x1AwhendH/du1=01whendH/du1<0, which can be written in the following compact form

u1∗= max0,min1,λ2x1∗x2∗−λ1x1∗A.

Similarly, by applying the conditions of optimality in Eq. (13), the characterization of the control variable u2 becomes

u2∗=0whendH/du2>0λ2x1x2BwhendH/du2=01whendH/du2<0, which can be written in the following compact form

u2∗= max0,min1,λ2x1∗x2∗B.

Then, the optimal solutions of the control variables are

u1∗,u2∗=max0,min1,λ2x1∗x2∗−λ1x1∗A,max0,min1,λ2x1∗x2∗B.

Hence the theorem completes the proof.

Results

Numerical simulations have been carried out by using MATLAB to investigate the findings of the rice-pest model (S4) and rice-pest-control model Eq. (7). This section aims to illustrate the dynamic change of the annual production of rice and the growth of the rice pests before and after adopting the control methods. In this case, the numerical values of parameters used in the simulations are taken from Table 1 and the initial conditions of the state variables chosen are x10 = 4.679, x20 = 0.05085. Here, the term “x 10” presents the initial annual production of rice in metric tons per hectare area (Mt/h) and the term “x 20” presents the initial growth rate of pests (for more details, please look at Supplemental Information 3). We consider 12 months (1 year) for the simulations. MATLAB codes are provided in Supplemental Information 4.

Table 1 A description of parameters used in this study with numerical values.

All of these are secondary parameters (derived/estimated or collected from other sources), two of which are estimated. We have conducted statistical analysis for parametric estimation after collecting and observing the corresponding data collected from different research (FAO, 2021; Tsuruishi, 2003; Vargas & Nishida, 1980). For more details, please look at Supplemental Information 3.

Symbol	Values	Descriptions	
α 1	4.679 metric tons/hector	Reproduction rate of rice	
α 2	94.915%	Mortality rate of the pest population	
β 1	37% of the total production	Losing rate of rice due to consumption of pests	
β 2	37% of the total production	Consumption rate of the pest population	
d 1	10% of the total production	Natural decrease rate of rice	
d 2	5.085%	Death rate of pests due to natural causes	

Numerical investigations of the rice-pest system (S4)

Both the time series and phase portrait of the considered dynamics are shown in Fig. 4. Initially, the growth of the rice plants sharply increases due to low numbers in the pest population. When the pest population increases by getting sufficient food, the growth of the rice plants is hampered so that eventually the pest population subdues (which in turn allows rice to grow again) (Vargas & Nishida, 1980; IPM, 2021). The phase portrait is shown in Fig. 4B spiralling out to a stable equilibrium at x ˆ1t,x ˆ2t=4.12,11.54 (Dym, 2004; Youssef & Raffoul, 2022).

Figure 4 Time series and phase portrait of the rice-pest system (S4).

(A) Time series of annual rice production and rice pest population and (B) phase portrait of the rice-pest system (S4) with the solution (4.12, 11.54).

Figures 5A and 5B show (i) when the consumption rate β2 of the pest population is reduced from 37% to 25%, the density of pest population decreases so that the rice is increasing again (Vargas & Nishida, 1980; IPM, 2021). Similarly, in (ii): when the consumption rate declines from 25% to 15%, the density of pests declines sharper than before, and the rice grows better again (Oerke et al., 1994; IPM, 2021).

Figure 5 Time series of (A) annual rice production, and (B) rice pest population for different consumption rates of the pests.

According to the result of Fig. 4A, the annual production of rice approaches 4.12 Metric tons/hector (Mt/h) whereas the growth rate of pests approaches 11.54%. The changes in the growth of rice for different consumption rates of pests are described in Fig. 5 which let us conclude that the production of rice can be increased if the consumption rate of pests is controlled.

Numerical investigations of the rice-pest-control system (Eq. (7))

When controls are applied to the rice-pest model (S4), the results change. The results of the rice-pest-control system Eq. (7) are described for the following three scenarios designed on the efficacy of the control variable u1 (cultural methods) and u2 (chemical controls): (i) u1 = 0.2 ∈ (0, 1] and u2 = 0, (ii) u1 = 0 and u2 = 0.26 ∈ (0, 1], and (iii) u1 = 0.1 ∈ (0, 1] and u2 = 0.2 ∈ [0, 1]. The simulations are performed distinguishing the cases “without control” and “with control”. Here, “without control” represents the results of the rice-pest system (S4) meaning there is no control strategy (u1 = 0, u2 = 0). The applications of controls are restricted by decision model Eq. (4) whereas pesticides are only applied in emergency situations to reduce the adverse effects of pesticides. Since the applications of pesticides are decided on the pest density, cultivation conditions and environmental factors, the decisions for the application of pesticides can vary over time and situations. Therefore, the application of u 2 can be applied just only once or several times, or maybe totally avoided.

(i) When only the cultural methods are implemented to the system as a control variable, the annual losses of rice decline from 41.05 to 38.39 MMT approximately, at the same time, the production rate of rice increases from 4.12 to 8.81 Mt/h approximately (Peshin & Dhawan, 2009; IPM, 2021; Milligan et al., 2016) which are shown in Fig. 6A. As a result of adopting cultural methods, the density of rice pests declines from 11.54% to 10.79% approximately over one period due to e.g., the enhancement of soil and rotation of crops (Peshin & Dhawan, 2009; IPM, 2021; Milligan et al., 2016) as shown in Fig. 6B. Figure 6C shows that the system under the application of cultural methods converges to the stable equilibrium point at x ˆ1t,x ˆ2t=8.81,10.79.

(ii) When only the chemical control is implemented according to the condition expressed in Eq. (4), the pest population comes down from 11.54% to 9.17% approximately that’s just over one-fifth of the total population (Inao et al., 2008; Bhattacharyya & Bhattacharya, 2006) as shown in Fig. 7B. Because of the decreasing pest population, the annual losses of rice drop sharply from 41.05 to 32.62 MMT approximately with a decrease approximating 25%. As a result, the annual rice production rate substantially grows from 4.12 to 12.86 Mt/h approximately (Inao et al., 2008; Marinelli, 2005) which is shown in Fig. 7A. The system converges to a stable equilibrium point at x ˆ1t,x ˆ2t=12.86,9.17, presented in Fig. 7C.

(iii) When the cultural methods are used continuously, chemical controls are according to decision model Eq. (4) only applied in an emergency i.e., only when the pest population density crosses the action threshold, which leads in a consequence to a decrease of the pest population, e.g., as shown in Fig. 8B from 33.02% to 7.73% (Inao et al., 2008; Peshin & Dhawan, 2009; IPM, 2021; Milligan et al., 2016). As a result, the annual losses of rice dramatically fall from about 41.05 to 27.49 MMT. After simultaneously applying controls the annual rice production increases considerably and reaches 19.18 Mt/h (Inao et al., 2008; Peshin & Dhawan, 2009; Milligan et al., 2016) as presented in Fig. 8A thereby approaching the equilibrium point x ˆ1t,x ˆ2t=19.18,7.73 as shown in Fig. 8C. The phase plane (Fig. 8C) reveals that the system with two controls stabilizes faster than the system with only one control.

Figure 6 Time series of the annual production of rice and pest population when only cultural methods are adopted as a control strategy.

Time series of (A) annual rice production under control u1 only, (B) pest population under control u1 only, (C) phase portrait of the rice-pest-control system Eq. (7) when only u1 is adopted, where the solution is (8.81, 10.79).

Figure 7 Time series of the annual production of rice and pest population when only chemical controls are adopted as a control strategy.

Time series of (A) annual production of rice under control u2 only, (B) pest population under control u2 only, (C) phase portrait of the rice-pest-control system Eq. (7) when only u2 is adopted and the solution is (12.86, 9.17).

Figure 8 Time series of the annual production of rice and pest population when both controls, cultural methods and chemical controls are implemented.

Time series of (A) annual production of rice under controls u1 and u2, (B) pest population under u1 and u2, (C) phase portrait of the rice-pest-control system Eq. (7) when both u1 and u2 are implemented, where the solution is (19.18, 7.73).

Next, we made a numerical comparison to analyse the results of all situations. The dynamic changes in the growth of the state variables in the three scenarios are shown in Fig. 9. As can be seen from Fig. 9, the annual production of rice increases significantly under both control strategies instead of just one, in contrast, the rice pests decrease dramatically. It also shows that cultural methods can control the density of the pest population, but chemical controls are comparatively more effective. From scenarios (i) to (iii), it is concluded that scenario (iii) is the best strategy to increase the annual rice production and reduce the density of rice pests.

Figure 9 A comparison between scenarios (i) to (iii).

Here, (A) The time series of the annual production of rice; (B) the time series of pest populations under three different scenarios.

Significance of β as a transcritical bifurcation parameter

Furthermore, the rice-pest model (S4) is experienced a transcritical bifurcation analysis and the dimensionless rice-pest model (S16) has been numerically investigated for the variation in the growth of pest populations (β) by employing MATLAB, where the dimensionless parameters α = 1 and γ = 0.001 (for more details, see Supplemental Information 2).

For β < 1, there is no intersection between the rice isocline and pests isocline as represented in Fig. 10A. In this case, the system (S16) experiences unstable and no equilibrium point (Banerjee & Petrovskii, 2011). For β = 1, the isoclines of rice and pests intersect at the pest-free equilibrium point E1(1, 0) as shown in Fig. 10B. For β > 1, the system (S16) experiences an interior point E∗(x∗, y∗) between the origin (0, 0) and E1(1, 0) as represented in Fig. 10C (Perko, 2000; Sen, Banerjee & Morozov, 2012). The nature of the system (S16) is illustrated at the interior equilibrium point E∗(x∗, y∗) for the different values of β > 1 which is described in Fig. 11. The rice-pest system (S16) is stable at the interior critical point E∗ for the bifurcation parameter β ∈ (1, 11]; e.g., the system is stable for choices of β = 2 and β = 10.9 as shown in Figs. 11A and 11B, respectively. For β = 11.1, there is a stable limit cycle at E∗ as shown in Fig. 11C. The system experiences a limit cycle for the parameter β ∈ [11.1, 13.6] which remains at a steady state, as presented in Figs. 11D to 11F. The steady state becomes unsteady state for β = 13.7 and thus the sytem becomes unstable for all β > 13.6 (for more details, see Fig. S3). Hence, the maximum value of β is β∗ = 13.6. Moreover, the transcritical bifurcation diagram with respect to the bifurcation parameter β has been carried out, as represented in Fig. 12. The diagram also describes that the dimensionless rice-pest system (S16) is stable for β ∈ (1, 11] and experienes steady state limit cycle for β ∈ [11.1, 13.6] and gradually tends to an imbalance situation with the increase of the growth of the pest populations (β) and even the system will be defeated for high growth rate (β > 13.6). Hence the bifurcation analysis suggests to control the pest population for the sustainable management of the rice-pest system (S4), otherwise, the high pest populations will likely reduce the ‘rice’ response to zero.

Figure 10 Diagram for the isocline of the rice (vertical) and the pest’s isocline (inclined)

(A) β = 0.8; (B) β = 1; (C) β = 2, where α = 1 and γ = 0.001 remains same.

Figure 11 Phase plane describing the nature of the rice-pest system (S16) at the interior point for the variation of bifurcation parameter (β)

(A) β = 2, (B) β = 10.9, (C) β = 11.1, (D) β = 12, (E) β = 12.66, and (F) β = 13.6, where α = 1 and γ = 0.001 remain same.

Figure 12 Transcritical bifurcation diagram of the rice-pest system (S16) for the bifurcation parameter showing whether the system is stable, limit cycle and unstable.

The bifurcation diagram shows that the system (S16) is stable for β from 1 to 11, experiences limit cycle for β from 11.1 to 13.6, and unstable for β from 0 to <1 and for >13.6. The figure reveals that the rice-pest system (S16) is present within the acceptable thresholds of the pests population and is destroyed above and below the acceptable thresholds.

Discussions

We developed a novel decision model considering cultural methods and pesticides as two control techniques to determine when controls should be applied and/or stopped. Since pesticides are potentially toxic and unsafe with detrimental effects on the environment and nearby ecosystems, it is recommended to apply chemical control only in emergency situations. The decision model is activated only when the number of individuals of a pest population exceeds the acceptable threshold and the model becomes inactive only when the level of acceptable damage threshold reaches the maximum level, which means that the model becomes inactive when environmental pollution reaches the maximally acceptable level. This situation continues to recur until the end of cultivation. The acceptable thresholds of pest population can vary with the pest species. In this case, the pesticide application time can change for the change in the acceptable thresholds.

Cultural methods have the potential to control rice pests in agriculture sustainably and simultaneously increase the annual production of rice as shown in Fig. 6. However, the use of chemical controls can further improve the yearly rice yield as shown in Fig. 7. Comparing Figs. 6 and 7, chemical controls show to be more effective and easier to apply than using cultural controls, contributing to higher annual yield thus contributing to local and global food security. For cultural methods, the system can reduce the growth rate of rice pests by about 33% which reduces the annual losses by about one-third, as represented in Fig. 8.

The comparison of three scenarios (Fig. 9) finds that the best control strategy for increasing the annual rice production and controlling rice pests being the use of cultural methods until the end of cultivation with the potential to use chemical controls only in emergency situations. In this study, we assumed pesticides control pests when they are applied and once the effect of pesticides falls below a certain level, the effect is statistically insignificant for further reducing pests. The pest species population will take only minor, non-lethal damage. In this scenario, only cultural methods remain for control. However, there may persist certain memory effects within the individual insect, its population or the host plant, which could be considered in the mathematical model as a feedback mechanism. Also, the rotation of crops in cultivation may have a significant role in increasing rice production and controlling pest infestations simultaneously. Especially the effect of memory or residuals of pesticides might have an effect on crop yield and rice-pest dynamics. For the current study the rotation of crops refers to a single crop in one season and then a different crop in the following season and so forth. Moreover, the long-term effects of using chemicals even in smaller numbers is outstanding research–chemical control can bring short-term relief but may damage the ecosystem with time. Here, the inclusion of changing water quality of nearby aquatic ecosystems, such as rivers or canals, in the mathematical model as a feedback mechanism could be considered. Furthermore, the introduction of biological controls such as certain predators or parasites as alternatives to chemical controls would make sense to be studied in combination to have more control variables and a decision model aimed at increasing simultaneously productivity and sustainability.

The newly formulated decision model and rice-pest-control model showed for the first time that the annual global production of rice is increased by mitigating annual global losses through the reduction of adverse effects of pesticides on the environment and nearby ecosystems. Understanding the rice-pest system dynamics as demonstrated in this article will assist researchers in developing new methods and ideas to contribute to improving global food security.

Supplemental Information

Supplemental Information 1 Modelling of the rice-pest dynamic system and its biological control

Click here for additional data file.

Supplemental Information 2 Bifurcation analysis for the rice-pest system

Click here for additional data file.

Supplemental Information 3 Data analysis

Click here for additional data file.

Supplemental Information 4 MATLAB scripts

Click here for additional data file.

Additional Information and Declarations

Competing Interests

Author Contributions

Data Availability

The authors declare there are no competing interests.

Sajib Mandal analyzed the data, prepared figures and/or tables, authored or reviewed drafts of the article, conceptualisation, methodology, software, validation, formal analysis, investigation, writing - original draft, and approved the final draft.

Sebastian Oberst analyzed the data, prepared figures and/or tables, authored or reviewed drafts of the article, conceptualisation, methodology, investigation, validation, writing - review and editing, and approved the final draft.

Md. Haider Ali Biswas analyzed the data, prepared figures and/or tables, authored or reviewed drafts of the article, conceptualisation, methodology, validation, resources, supervision, project administration, and approved the final draft.

Md. Sirajul Islam analyzed the data, prepared figures and/or tables, authored or reviewed drafts of the article, conceptualisation, supervision, and approved the final draft.

The following information was supplied regarding data availability:

The rice-pest model and the transcritical bifurcation analysis of the rice-pest model, the data, and the MATLAB codes used for carrying out all numerical simulations are available in the Supplementary Files.

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
