# Peer review of "Dynamic analysis and control of a rice-pest system under transcritical bifurcations"

_PeerJ, doi:10.7717/peerj.16083_

## Round 0.1 · original submission · Major Revisions

Dear Dr Mandal,

Thank you for submitting your text intended for publication, which I consider interesting for the Journal and me.

The referees highlighted several points requiring clarification and better formal treatment.

The comments clearly describe the referees' intentions to ameliorate the text, suggesting the completion of missing but relevant portions of descriptive and conclusive phrases.

Please consider the comments about self-contradictory statements and suggestions for a better phenomena treatment.

Even the real in-field rice crop interval deserves the proper attention.

From my point of view, I wish to suggest detailing the damaging pest features.
Because of the general opinion, I cannot accept the submission as it is. Permit me to suggest a prompt resubmission as soon as you incorporate the comments in the new submission version.

Thank you very much,
Francesco

Reviewer 1 ·

Basic reporting

no comment

Experimental design

no comment

Validity of the findings

no comment

Additional comments

Article reference: PeerJ #77522

Review of the manuscript “Dynamic analysis and control of a rice-pest system under transcritical bifurcations”, by Sajib Mandal et al.


The aim of the present work is the developing of a control model useful in the rice production, that is generally suffering from pests.
In the Introduction section the Authors give a brief nevertheless exhaustive overview of the pests control in agriculture, stating that it can be carried out through both the chemical control (i.e. pesticides) and the cultural methods and biological control. Beside, they stress the importance of optimized integrated pests management (IPM) strategies, that simultaneously take into account for different kinds of control, and they report the state of the art on this topic. The discussion is supported by many cited references.
In this framework they develop and propose their own rice-pest-control model, starting from the definition of a proper decision model, in which the two ‘controls’ are the cultural methods and the chemical control, to end in the suggestion of an optimal control problem (OCP). One of the goodness of their formulation is that the use of pesticides is limited to emergency situations.

The paper is really interesting and generally well written, however some points should be clarified before publication.
The page numeration, in the queries below, starts from the first page of the file created by the Journal, so that page 5 corresponds to page 1 of the original manuscript.

1 – page 6, rows 31-32.
A sentence or almost a word to join, from the logical-discursive point of view, the OCP definition to the Lotka-Volterra model definition would be beneficial, even if not mandatory.

2 – page 6, rows 32 and 35.
Once u(t) is named a control variable and once it is named a control function, this should be either corrected (and kept uniform through the paper) or clarified.

3 – page 6, row 38.
Is “a.e.” a sort of typo (maybe due to the transformation of the original file in a .pdf one)? Does it means ‘for all’?

4 – page 7, rows 33 and 36.
The Authors should clarify if D is a damage function with respect to the environment (as it seems by “denote the function of the nearest ecosystem which is potentially subject to pesticide damage”) or to the rice crop (as it seems by “… D, is increased in the rise of x2 and E but decreased in the application of u2…”).

5 – page 7, row 35 and page 12 row 32.
It is written: “E =1 denoting that the pollution reaches the most acceptable level (E=0 denotes zero pollusion).”
However E=1 is less acceptable than E=0 (that denotes zero pollusion), do the Authors mean “E =1 denoting that the pollution reaches the highest acceptable level”?
Analogously in the Conclusion Section: “… when environmental pollution reaches the most acceptable level.”

6 – page 8, row 4 and Fig.2.
In the Fig.2 it should be clarified how the model adopted causes the pests number x2 decreasing when D=1 and the pesticide use is stopped (u2=0).

7 – page 8, rows 21 and 27.
It is not clear which term of Eq.(6) correspond to u1x1x2 (to give an example, the term F(D,t)x1(t)x2(t), due to Eq.(4), corresponds to u2x1x2 mentioned in row 24).

8 – page 8, row 35.
As in the Supplementary Material, it would be good to mention that the rice-pests system (7) has a Lotka-Volterra-like model form.

9 – page 9, row 6.
In Eq.(9) it is written J(u1,u2); the Authors should check if it is a typo in place of J(x,u) (see Eq.(10) at row 8).

Reviewer 2 ·

Basic reporting

The authors propose an optimal control model to mitigate annual losses of rice by controlling insect pest infestations using traditional cultural method and pesticides taking the adverse effects of pesticides on the environment and nearby ecosystems into consideration. The proposed model uses a threshold function to switch between the traditional cultural method and use of pesticides or to use both in order to suppress the growth of pest population and enhance the growth rice. The problem itself quite interesting. However, some assumptions in the model contradict with the explanation in the text and hence, these terms require further attention/modification. Bifurcation analysis needs further clarification. In addition, several parts of the manuscript require re-writing.

Experimental design

1. The function D(x_2,u_2,E)∈[0,1] characterizes the threshold for stopping the use of pesticides. E also represents a similar threshold. I was wondering if it is still essential to use E while already D defines a similar threshold. Besides, the concept of the function appears to be a little unclear to me. Without using pesticides (u_2), the value of x_2 alone can trigger the value of D that may ask the system to stop using pesticides when its threshold is attained, which clearly goes against the objective of the study. Nowhere in the text, it has been said what explicit functional form of D(x_2,u_2,E) was used for numerical simulations. Having seen the explicit form could clear the aforesaid facts and if still not, then the functional form needs to be redefined.

2. Right before introducing Eq. (5) it’s been said, “When cultural methods (u_1) are applied during cultivation, the production rate of rice increases rapidly. Since the use of cultural methods does not depend on the density of rice pests”, and using this assumption the term u_1 x_1 has been added in Eq. (5). To my understanding, if this assumption is true then there will be a very little room left for fitting the concept of using pesticides when needed. Natural method alone will only be effective in increasing the rice production significantly when the pesticides (x_2) level will remain under a tolerable threshold. It is indeed true that cultural method could be effective in suppressing the growth pest population and hence alternatively increase the production of rice to some extent, and that has already been incorporated in the first term (α-β_2 x_2 ) x_1 of Eq. (5) (less x_2 population implies more production of grain). The effectiveness of cultural method should be defined considering the role of existing pesticides level. I therefore believe that the term u_1 x_1 should be removed or amended by taking the role x_2 level in neutralizing the effort paid by u_1.

3. When x_2 level is below the threshold A_T, the term -F(D,t) x_1 x_2 contradicts with the definition of cultural method as stated in the middle of the first paragraph in Introduction section. When x_2<A_T, the elimination of pesticides would be cultural method dominated and, in that case, it would be more appropriate if this term is defined as -F(D,t) x_2 since the rotation of crops essentially means there will be one crop in one season (probably rice) and a different crop in next season or so. It is noteworthy to mention that, the authors simulate the model for one year and the required time for completing the cultivation of rice is not usually more than 4 to 5 months. Note that this contradiction does not get any better while considering the removal of diseased plant which is another approach in cultural method. So, when x_2<A_T, the reduction in pest population due to the use of cultural method could more appropriately be modelled by using -F(D,t) x_2 instead of -F(D,t) x_1 x_2.

Validity of the findings

1. Figure1 depicts a graph that demonstrate the annual losses of rice production. Undoubtedly, this graph motivates the importance of this study though it does not confirm that these yearly losses are not caused by pest infection only, could be one of the reasons behind this continuous production fall over the years. Deforestation and urbanization are occurring at an alarming rate around the world. In addition, compared to the other crops, rice is less profitable so people around the world are slowly inclining towards cultivating those profitable crops rather cultivating rice. These facts should be acknowledged where appropriate. Acknowledgment of these facts does not reduce the importance of this study whatsoever.

2. In bifurcation analysis, it’s been said when β>31, there is a limit cycle around co-existing equilibrium and the system remains in oscillation with respect to time but the graph shows that those oscillations are just damped oscillation where the system oscillates a bit before settling down to it equilibrium value. The existence of limit cycle means the stable co-existing equilibrium changes its stability criterion from being stable to unstable and a stable limit cycle emerges through Hopf bifurcation when β crosses 31. If that’s the case, is it possible to conduct an analysis, either theoretical or numerical, supporting the existence of Hopf bifurcation and a stable limit cycle that is born through this Hopf bifurcation?

Additional comments

1. Define the acronym MMT where it is used for the first time—Line 9.
2. The word “arise” does not seem to be fit in the context of the sentence —Line 13.
3. The first three paragraphs in Subsection 3.1 would fit best in the Introduction section. If they are already been said in the Introduction section, then there is no use of repeating the same thing here again.
4. In —Line 35, “most acceptable” sounds like the most awaited and desirable level of x_2, which I believe not meant to be said.
5. The time series solution of the variables clearly provides all the required details about the dynamics of the system. Do the phase planes represent anything extra about the dynamics that cannot be seen the time series solution? If so, please mention otherwise it would be wise to remove them.

Annotated reviews are not available for download in order to protect the identity of reviewers who chose to remain anonymous.

Reviewer 3 ·

Basic reporting

There are repetitive sentences in the abstract. Lines 11-12 are similar to lines 18-19.

Also, consider rewriting the sentences for lines 19-22, as the sentence meaning are not concise.

Especially, the authors mentioned about "acceptable threshold" in line 22. However, no such threshold was reported in the results.

Line 33 on page 1 and line 13 on page 3 define cultural methods. Please remove repetitive parts.

Similarly, chemical control was defined twice, once in line 28 of page 1 and again in line 15 of page 3.

Please consider a professional English editing service for the improvement of your writing.

Experimental design

The manuscript belongs to the scope of the journal. The analysis does not completely answer the research question. The solution technique for the optimal control problem was not described.


In lines 6 to 10 of page 2, the authors mention "minimum adversity of pesticides on the environment". But the results do not show any measure of adverse effect on the environment, and neither the model reflects this.

Line 11-12, page 7: The authors mentioned using ODE45 to solve the problem. However, ODE45 cannot solve optimal control models.

Line 15, page 7: The parameter values are quite strange. The authors choose x_{10}=4.679 and x_{20} = 0.05085, where x_1 is the annual production of rice and x_2 is the density of the pest. The values are impractical, as according to the data shown in figure 1, annual production was above 100 MMT.

Results have been shown for different control strategies in figures 6-9, but corresponding graphs of u_1 and u_2 are not shown. In addition, Figure 6-9(c) is not necessary to show the equilibrium, as the equilibrium is evident from corresponding figures (a) and (b).

Table A1: \alpha_1 is the reproduction rate of rice which is not the same as production per unit area 4.679 MMT/ha.

Table A2: Why did the authors consider two different paste and took their average mortality rate?
Several pastes may survive on a single paddy plant. So, \beta_1 and \beta_2 cannot be the same. If the authors claim it to be the same, please provide a specific reference (not only the book name, please provide the page number).

Please check the sentence in lines 11 to 13 on page 2. Pontryagin's maximum principle is not for varifying optimal control problems.

Line 16, page 2: The authors mentioned "d_1 presents the decrease rate due to intraspecific competition", But in Figure S1, the authors mentioned d_1 as a natural decrease. Which definition is correct? Does the estimate in Supplementary C agree with this definition? Please check the same issue for d_2.

Validity of the findings

The results, numerical techniques, and parameter choice have several issues that hinder the assessment of the novelty.

Figure resolutions are poor.

Additional comments

Though the authors attempted to address an important research question, it lacks an adequate presentation of methods and results and proper parameterization.

---

## Round 0.2 · accepted · Accept

The previous Academic Editor is not available and so I have taken over handling this submission.

The reviewers' concerns have been addressed. It now looks good. I recommend it for publication.

Reviewer 1 ·

Basic reporting

no comment

Experimental design

no comment

Validity of the findings

no comment

Additional comments

The Authors properly addressed the queries

Reviewer 2 ·

Basic reporting

The updated manuscript is much improved over the previous versions. I am pleased that the authors have found my comments useful and have managed to improve their paper following my suggestions. I recommend that the manuscript now be accepted for publication.

Experimental design

The authors have addressed my concerns appropriately.

Validity of the findings

New analyses have been added to strengthen existing ones following my comments.